# Improving the Autofluorescence of *Lophira alata* Woody Cells via the Removal of Extractives

**DOI:** 10.3390/polym15153269

**Published:** 2023-08-01

**Authors:** Zhaoyang Yu, Dongnian Xu, Jinbo Hu, Shanshan Chang, Gonggang Liu, Qiongtao Huang, Jin Han, Ting Li, Yuan Liu, Xiaodong (Alice) Wang

**Affiliations:** 1College of Materials Science and Engineering, Central South University of Forestry and Technology, Changsha 410004, Chinaliugonggang@gmail.com (G.L.); hanjin@csuft.edu.cn (J.H.);; 2Department of Research and Development Center, Yihua Lifestyle Technology Co., Ltd., Shantou 515834, China; 3Hunan Taohuajiang Bamboo Science & Technology Co., Ltd., Taojiang 413400, China; 4Department of Wood and Forest Sciences, Laval University, Quebec, QC G1V 0A6, Canada

**Keywords:** fluorescence quenching, extractives, lignin, spectrometer

## Abstract

The autofluorescence phenomenon is an inherent characteristic of lignified cells. However, in the case of *Lophira alata* (*L. alata*), the autofluorescence is nearly imperceptible during occasional fluorescence observations. The aim of this study is to investigate the mechanism behind the quenching of lignin’s autofluorescence in *L. alata* by conducting associated experiments. Notably, the autofluorescence image of *L. alata* observed using optical microscopy appears to be quite indistinct. Abundant extractives are found in the longitudinal parenchyma, fibers, and vessels of *L. alata*. Remarkably, when subjected to a benzene–alcohol extraction treatment, the autofluorescence of *L. alata* becomes progressively enhanced under a fluorescence microscope. Additionally, UV–Vis absorption spectra demonstrate that the extractives derived from *L. alata* exhibit strong light absorption within the wavelength range of 200–500 nm. This suggests that the abundant extractives in *L. alata* are probably responsible for the autofluorescence quenching observed in the cell walls. Moreover, the presence and quantity of these extractives have a significant impact on the fluorescence intensity of lignin in wood, resulting in a significant decrease therein. In future studies, it would be interesting to explore the role of complex compounds such as polyphenols or terpenoids, which are present in the abundant extractives, in interfering with the fluorescence quenching of lignin in *L. alata*.

## 1. Introduction

Autofluorescence is a widespread phenomenon resulting from the presence of fluorescent compounds in various cellular compartments. The primary advantage of fluorescence microscopy methods lies in their ability to provide a colorful visualization of xylem, wherein plant samples can exhibit strong green fluorescence in the xylem cells [1,2,3]. Additionally, the utilization of staining methods to examine the microstructure of xylem cells under a light microscope can potentially lead to the degradation or modification of the woody slide [4,5]. It is widely recognized that the autofluorescence of lignin stems from the presence of conjugated bonds within its aromatic backbone [6,7]. This autofluorescence of lignin can aid in distinguishing lignified cells from tissues that lack fluorescence, such as hemicellulose and pectin, which are other components of the cell wall [8]. Notably, the application of lignin fluorescence technology holds promise for enhancing the visualization of growth rings and identifying false rings in various tropical species [9]. Moreover, it has the potential to assess cell wall modifications resulting from a range of biological, chemical, and physical treatments in the wood industry [2,4,5,8].

Azobe wood (*Lophira alata*), commonly known as African ekki wood, is predominantly found in the tropical rainforests of Africa, including the forests of Sierra Leone, Uganda, Gabon, and Sudan. In recent years, it has become a widely imported timber. This wood is highly sought after for its exceptional durability, making it a popular choice for marine applications and railway sleepers [10]. While *L. alata* is known for its toughness and longevity, it can raise challenges during the working process. However, its remarkable resistance to decay, particularly in marine environments, justifies the extra effort required [10]. Studies have reported that *L. alata* contains a rich variety of extractives, including polyphenols, tannins, flavonoids, alkaloids, and other compounds [11,12]. Notably, extracts derived from *L. alata* have demonstrated antibacterial effects and possess significant medicinal value [13]. Numerous extractives play a crucial role in shaping the properties and functions of wood. Many African woods, including *L. alata*, are known to possess extractives with antimicrobial and preservative properties. These substances inhibit the growth of bacteria, fungi, and other microorganisms, thereby extending the lifespan of the wood [14]. Additionally, the antioxidants present in the extractives contribute to reducing the aging process of the wood. Moreover, extractives have the potential to fill voids in the lignin, thereby increasing the density and durability of the wood [15].

The microstructure of *L. alata* is occasionally observed using fluorescence microscopy as depicted in Figure 1. Interestingly, it has been discovered that the lignin autofluorescence in *L. alata* is nearly imperceptible. It is speculated that the abundant extractives in Azobe wood could interfere with the lignin autofluorescence. In this study, we aim to investigate this accidental discovery by examining the heartwood and sapwood of *L. alata* using a fluorescence microscope. Additionally, the fiber morphology of *L. alata* is characterized by means of light microscopy and scanning electron microscopy (SEM). To further understand the phenomenon, the diffuse reflectance spectra and UV–Vis absorption spectra are applied to respectively investigate the light absorption and reflection in relation to both the wood and its extractives. In addition, the hypothesis that the extraction process enhances the visibility of wood autofluorescence is verified. By examining the microstructure and comparing the fluorescence properties of the extracted wood, the aim is to reveal the effect of the extractives on the autofluorescence quenching phenomenon in *L. alata.* This research contributes to the field of wood science by providing new insights into the mechanisms underlying the spontaneous fluorescence quenching phenomenon in *L. alata*. It also establishes a foundation for further applications that leverage the unique fluorescence properties of wood. Moreover, this study is of significant importance in unraveling the distinct characteristics of African wood species and effectively harnessing their potential, e.g., in applications for optical material.

## 2. Materials and Methods

### 2.1. Materials

The *L. alata* was grown in the tree farm of Yihua Lifestyle Technology Co., Ltd. (Shantou, China) in Gabon. One tree with a straight trunk and a diameter of approximately 110 cm at breast height was sampled for this study. Wood discs of approximately 5 cm thickness were collected at a height of 1.2 m. Subsequently, the collected discs were stored for over 6 months to allow for moisture content adjustment to approximately 12%. After air-drying, wood samples were obtained and processed into test samples of varying dimensions. Wood powders with ~200 mesh sizes were prepared for the determination of the diffuse reflection and the UV–Vis absorption spectrum. As a comparison of fluorescence observation, a poplar tree (*Populus deltoides*) from the Jiaozuo experimental field in Henan (Jiaozuo, China) was sampled.

### 2.2. Autofluorescence Experiment

Heartwood and sapwood samples of 10 × 5 × 5 mm^3^ (longitudinal × radial × tangential) were prepared, respectively. Meanwhile, wood blocks of the poplar wood were prepared for comparative observation. The samples were softened in boiling water until they could be easily sliced off with a small blade [16]. Slices of 15 μm thick cross sections were cut with a sledge microtome (G.S.L.1, WSL, Zurich, Switzerland), and placed between a microscopy glass slide and a coverslip [3]. Then, the thin slices were observed under an optical microscope (Eclipse Ci-L, Nikon, Tokyo, Japan) with fluorescence mode. The wavelengths of fluorescence emissions were UV light (emission wavelength: 330–380 nm), blue light (450–490 nm), and green light (510–560 nm), respectively [17].

### 2.3. Diffuse Reflection Spectra and UV–Vis Absorption Spectra

The diffuse reflection spectra were recorded on a UV–Vis–NIR spectrometer (Lambda 750, PerkinElmer, Waltham, MA, USA) within a range of 200 to 800 nm using a diffuse reflectance regime. The sample for this test was wood powder of 200 mesh sizes. 

The UV–Vis absorption spectra were measured using a UV–Vis spectrophotometer (N5000, PerkinElmer, Waltham, MA, USA) in the range 200 to 800 nm. 0.1 g and 1 g of air-dried wood powders (200 mesh) of *L. alata* were extracted in a benzene-ethanol mixture solution (benzene: 95% alcohol = 2:1) for 8 h in a water bath at a temperature of 78 °C, respectively. The ratio of solvent volume to wood powder weight was 30 mL/g. Then, they were placed in a 100 mL hydrothermal reactor and underwent a hydrothermal carbonization treatment at 140 °C for 24 h, respectively [18]. The solution might have contained some lignin after the hydrothermal treatment, so 0.1 g of commercial lignin solution was prepared as a comparison. The supernatant was taken from the filtrated solution to obtain the UV–Vis absorption spectra.

### 2.4. Macrostructural Observation

Wood samples of 50 × 50 × 30 mm^3^ (L × R × T) were cut for macrostructure observation. A sharp blade was used to trim their planes to ensure their flatness and a smooth surface. The macrostructure was observed using stereomicroscopy (MZS0745, Evident Corporation, Tokyo, Japan). The macrostructure analyzed was mainly based in the standard of ISO 3129:2019 [19]. The main macroscopic structural features, including color, smell, the size and distribution of vessels, rays, and grains, were examined. 

### 2.5. Microstructural and SEM Observation

Test samples of 10 × 5 × 5 mm^3^ (L × R × T) were prepared for microstructural observation. Slices of 15 μm thickness were cut from softened samples, including cross, tangential, and radial sections. After staining with 1% safranine O solution and dehydration, the sections were fixed on the slide with neutral quick-drying glue and observed using an optical microscope (Eclipse Ci-L, Nikon, Tokyo, Japan). Thirty groups of fibers were randomly selected in cross section, and their double wall thickness and cell lumen diameter were measured using image analysis software (ImageJ 1.50i). Meanwhile, 12 cross sections were observed and photographed to measure the proportion of each major cell type. 

Small blocks with dimensions of 5 × 5 × 5 mm^3^ (L × R × T) were trimmed with a new single-edged razor blade for each surface [17]. Before testing, the blocks were coated with a thin layer of gold under vacuum. Then, samples were observed using scanning electron microscopy (SEM, Sigma 300, ZEISS, Jena, Germany) under a secondary electron detector. 

## 3. Results

### 3.1. Autofluorescence Experiment

The fluorescence observation images of the heartwood and sapwood of *L. alata* are presented in Figure 1. Slightly fluorescent signals were observed using fluorescence microscopy in both the sapwood and the heartwood (Figure 1). Specifically, their autofluorescence signal was remarkably weak, rendering it nearly invisible, particularly in the heartwood. Only faint fluorescence signals were vaguely discernible in a few fiber cell walls. Furthermore, distinct ribbon cells of longitudinal parenchyma were observed in both the heartwood and the sapwood (Figure 1). Notably, black extractives were more abundant in the longitudinal parenchyma cells of the heartwood (Figure 1A). As a comparison, the cell walls exhibited noticeable blue, green, and red fluorescence under UV, blue light, and green light excitation, respectively (Appendix A, see in Appendix A). The fluorescence images of the poplar wood clearly depicted the outlines of fiber cells. Therefore, further investigation was undertaken to understand the underlying reasons for the near invisibility of *L. alata*’s autofluorescence.

### 3.2. Diffuse Reflection Spectra and UV–Vis Absorption Spectra

To investigate the spectral absorption and reflection properties of *L. alata*, diffuse reflection spectra and UV–Vis absorption spectra were conducted, and the results are presented in Figure 2. The spectral analysis revealed a significant absorption of light by *L. alata* within the wavelength range of 200–500 nm. However, beyond the 500 nm wavelength, there was a significant increase in the reflection of light, based on the Kubelka-Munk formula [20]:(1)$$K/S={\left(1-R\right)}^{2}/2R$$
where *R* represents the diffuse reflectance of the layer relative to the standard, *K* is the absorption coefficient, and *S* is the scattering coefficient. Since wood is a good light absorber, the K-M theory is widely applied to determine its light absorption [21,22]. As illustrated in Figure 2, the diffuse reflectance spectra were transformed into an absorption spectrum. The peak of the *K/S* spectral line appears at 334 nm, when the *K/S* value exceeds 20, indicating that the material exhibits an exceptionally strong light absorption capacity within this range [22]. Notably, *L. alata* demonstrated varying light reaction abilities across different wavelength ranges. Specifically, within the lower wavelength range of light (particularly below 500 nm), *L. alata* exhibited a robust light absorption capacity alongside a relatively weak light reflection ability.

The UV–Vis absorption spectra of the extracted solutions and pure lignin are depicted in Figure 3. The spectrum of pure lignin exhibits a distinct absorption peak at 280 nm, which is characteristic of lignin absorption [23]. Comparing the UV–Vis absorption spectra of commercial pure lignin, the extractives of *L. alata* demonstrate strong light absorption within the wavelength range of 200–500 nm. Notably, the UV–Vis absorption spectra of the extractives obtained from 1 g of wood powder exhibit significantly higher absorption compared to those obtained from 0.1 g. Moreover, the extractives derived from 1 g of wood powder have a strong absorption peak at 235 nm, along with slightly weaker peaks at approximately 360 nm and 391 nm. These peaks indicate a significant absorption of light by the extractives of *L. alata* at these specific positions under the same extraction conditions. Notably, the intensity of absorption bands within the 200–500 nm range decreases when the wood powder used for extraction is reduced from 1 g to 0.1 g. Meanwhile, the absorption peaks at 360 nm and 391 nm may be attributed to the presence of lignin polyphenols and ketones in the wood [24,25]. The characteristic absorption peak of lignin at 280 nm is observable in the spectrum of pure lignin. This peak is primarily attributed to the π-π* transition within the aromatic ring and serves as an indication of the presence of lignin. Compared to the absorption spectra of pure lignin, there is almost no lignin in the extracted solutions, and the extractive components demonstrate a greater light absorption capacity. In summary, the extractives from *L. alata* exhibit significant light absorption within the range of 200–500 nm.

### 3.3. Macroscopic Observation

As shown in Figure 4, a dark brown color is present in the heartwood of *L. alata*. Meanwhile, the growth rings are indistinct, and the wood exhibits diffuse-porous characteristics. Most of the vessel lumens are filled with rich extractives (Figure 4A-c,B-c), which become more obvious in radial (Figure 4C) and tangential (Figure 4D) sections. Under the stereomicroscope, pores contain light-colored mineral deposits that form small but conspicuous streaks throughout the wood. Both the longitudinal parenchyma (Figure 4A-b,B-b) and wood rays are visible. The wood rays are narrow and densely distributed. The longitudinal parenchyma tissue resembles a thin white line perpendicular to the wood ray (Figure 4B). With the dense and hard texture of *L. alata*, the dark color also allows it to absorb light easily.

### 3.4. Microscopic Observation

The microstructure images of *L. alata* are displayed in Figure 4. Upon observing the cross section (Figure 5A), it is evident that the vessels have an oval shape with multiple pores (mostly two) and occasionally solitary pores. Some vessels also contain extractives (Figure 5e) within their interior. The perforation plate between vessels exhibits inclined simple perforations, while spiral thickening is absent. Notably, a significant abundance of longitudinal parenchyma is observed in the wood cross sections, presenting as a string-like apotracheal banded pattern with a width of 3 to 4 cells (Figure 5A-c,B). The fibers of *L. alata* possess an elliptical shape in the cross section, and their cell walls are thick (Figure 5A,B). 

The measurement results of the cross section, as indicated in Table 1, reveal a fiber cell wall-to-lumen ratio of 5.9 and a cavity-to-diameter ratio of merely 0.14. The fiber cavities are small, while the cell walls are exceptionally thick. The rays are made up of homogeneous, uniseriate, and multiseriate cells (typically 2 to 3 cells wide) and are composed of procumbent cells (Figure 5D,E). Most of the cells in the wood rays are filled with extractives and gum (Figure 5D-b). Microscopic observation clearly demonstrates the presence of abundant dark extractives in the longitudinal parenchyma and rays. 

In the cross section, the main cell tissues include vessels, fibers, rays, and longitudinal parenchyma. The results from measuring the proportions of the four tissues are shown in Table 2. Longitudinal parenchyma and wood rays containing abundant extractives account for 14.4% and 17.2%, respectively. The predominant cell tissue of *L. alata*, which has a dense and hard texture, accounts for 58.2% of the fibers. It may be speculated that the rich extractives and extremely thick cell walls of the fibers could explain the wood properties of *L. alata*.

As depicted in Figure 6, the SEM observations provide a clearer view of the fibrous structure of *L. alata*. The fibers exhibit thick cell walls, while the cell lumens are either absent or very narrow (Figure 6A). The vessels exhibit a drum-like or cylindrical shape in both radial and tangential sections, with some containing abundant extractives within their cell lumens (Figure 6B,C). The intervascular pits appear predominantly elliptic or slit-shaped, with a diameter of approximately 2–3 μm (Figure 6D). Occasional vestured pits can also be observed (Figure 6D-d). Abundant extractives are observed within the lumens of the longitudinal parenchyma (Figure 6E) [26]. Additionally, a substantial amount of extractives fills the wood ray cells (Figure 6F). Moreover, numerous pits are visible in the cell walls of the longitudinal parenchyma (Figure 6F-e). Microscopic and SEM observations confirm the presence of a significant quantity of extractives in the longitudinal parenchyma and wood rays, effectively filling the cell lumens. Furthermore, the observations highlight that the wood is rich in extractives and the fibers exhibit a near-solid structure. This phenomenon may enhance the light absorption capabilities of the wood while impeding light transmission.

## 4. Discussion

Two possible conditions are generally recognized as necessary for a substance to exhibit fluorescence: (1) the absorption of relatively low energy photons undergoing multiple invariant transitions, and (2) a fluorophore containing an unsaturated bond in the structure of the substance [27]. Based on these conditions, it is hypothesized that there are two main reasons for the decrease in fluorescence observed in *L. alata*. The first reason could be attributed to the absence of fluorescent luminescent factors. Secondly, it may be a result of either direct energy transfer between fluorescent and quenching molecules or the absorption of fluorescence emission, known as fluorescence quenching [28,29,30,31].

To investigate the factors contributing to the reduced fluorescence in the fibrous cells of *L. alata*, including the high-content extractives, anatomical features such as fiber morphology and tissue ratios were investigated. As presented in Table 1, the fiber cell wall-to-lumen ratio was found to be 6.1, while the cavity-to-diameter ratio was only 0.14. Microscopic and SEM observations revealed that *L. alata* exhibited extremely thick cell walls and nearly solid cells. Moreover, the predominant cell tissues in *L. alata* were found to be fibers, accounting for 58.2%, according to the measurement results in Table 2. In fact, it has been proved that some obscure photography of partial cells has been obtained using a light microscope equipped with a camera. It is well known that wood cell walls are made up of cellulose, hemicellulose, and lignin [32]. As the main component of the cell wall of wood, lignin exhibits a broad range of fluorescence emission and can be excited by both UV and visible light [33]. Analysis of the physicochemical structures and fluorescence properties of lignin suggests that the aggregation-induced conjugation of phenylpropane units is the main source of visible emission in lignin, leading to the formation of different phenylpropane aggregates in lignin micelles, thus creating a multi-fluorophore system [34,35]. Therefore, the microstructure of lignin plays a crucial role in its complex fluorescence properties due to fluorophore interaction and aggregation behavior. However, the autofluorescence of *L. alata* is nearly invisible when observed using fluorescence microscopy (Figure 1). Simultaneously, the presence of lignin in the fiber cell walls indicates that the aforementioned reason (1) is not valid.

In Figure 5, the longitudinal parenchyma and wood rays are predominantly filled with dark extractives. These extractives are formed through cellular metabolism and biochemical reactions, accumulating in different regions of the wood cells [36]. The presence of abundant extractives in the longitudinal parenchyma (Figure 6E) and ray cells (Figure 6F) was further confirmed by means of SEM observations. The longitudinal parenchyma and rays constitute 31.6% of the tissue ratios (Table 2). Additionally, some pits within the vessel walls are also filled with extractives (Figure 6D).. Throughout the process of tree growth and accumulation, it is suggested that the plentiful extractives might gradually seep out from the parenchyma tissue and ray cells. These extractives could then penetrate specific vessels by passing through the pits in the cell walls. The presence of extractives in the vessels (Figure 6B,C) further supports this observation. The numerous pits (Figure 6D,F) within the cell walls of vessels and rays provide a pathway for the penetration of extractives. Moreover, the rich extractives and extremely thick cell walls of fibers contribute to the high density and hardness observed in *L. alata*. The presence of these extractives in the wood cells is also responsible for the dark brown color observed in macrostructural observations (Figure 4). Hence, it can be hypothesized that the significant amounts of extractives in the wood contribute to the quenching of lignin fluorescence [37,38].

To investigate the impact of extractives on autofluorescence, benzene-ethanol extraction tests were conducted on wood slices. The slices, with a thickness of 15 μm, were subjected to benzene-alcohol extraction in a Soxhlet extractor at a water bath temperature of 78 °C (benzene:95% alcohol = 2:1) [39]. At intervals of 2 h, the slices were removed for fluorescence observation. As depicted in Figure 7, fluorescence gradually emerged in the wood following the benzene–ethanol extraction. Interestingly, a relatively distinct outline of the fiber cells became visible after 8 h. However, the fluorescence observation still failed to reveal the longitudinal parenchyma and wood rays. This suggests that the accumulation of abundant extractives in the longitudinal parenchyma and ray cells may be responsible for this phenomenon. Additionally, it is possible that benzene–alcohol extraction does not effectively remove the decades-old deposition within the parenchyma tissues, while the extraction of small quantities of extractives from the fiber cells may be relatively easier. Previous studies have indicated that wood extractives primarily consist of polyphenols, tannins, flavonoids, alkaloids, and other compounds [40]. Some of these chemical components within the extractives act as quenchants, interacting with fluorescent substances, particularly lignin fluorescence, leading to fluorescence quenching. When the fluorescent molecule is in the excited state, contact with the quenchants prevents the emission of fluorescent photons and causes the molecule to return directly to the ground state [41]. Consequently, the analysis and exploration of extractives should be further investigated in subsequent studies, employing techniques such as liquid chromatography–mass spectrometry (LC–MS), nuclear magnetic resonance (NMR), matrix-assisted laser desorption/ionization time of flight mass spectrometry (MALDI-TOF-MS), among others.

Previous studies have confirmed that the fluorescence emission of poplar fibers appears to correspond to the known distribution of syringyl-rich lignin in the fiber secondary wall [5,42]. It has also been demonstrated that the fluorescence intensity is not solely determined by the amount of lignin but is also influenced by the molecular environment and chemical structure [4,5]. Wood extractives are diverse chemical components that result from physiological activity or metabolic processes during tree growth [43]. The mechanisms underlying autofluorescence in wood are highly complex and easily influenced by these compounds. Based on the results of UV–Vis absorption spectra, the extractives from *L. alata* exhibit strong light absorption in the wavelength range of 200–500 nm. The presence of abundant extractives and their significant light absorbing capacity may contribute to fluorescence quenching [37,44], which could potentially explain the exceptional corrosion resistance, high durability, and remarkable density observed in this species. It is hypothesized that other tree species with abundant extractives may exhibit a similarly inconspicuous fluorescence observation.

## 5. Conclusions

In this foundational study, the autofluorescence of *L. alata* was extremely faint and nearly imperceptible when observed using a fluorescence microscope. Gradually, fluorescence can be shown more clearly in the fiber cell wall region due to the removal of certain organic extractives by means of benzene-alcohol extraction. Abundant extractives were observed in the longitudinal parenchyma and wood ray cells of *L. alata* through optical microscopy and SEM observation. Moreover, further investigation was warranted to understand the fluorescence-quenching phenomenon in the parenchymatous tissue and ray cells, where these rich extractives could potentially contribute to a more complex interaction. Additionally, the UV–Vis absorption spectra of the extracted solutions revealed an exceptionally high light absorption capacity of the extractives in *L. alata*. These findings could suggest a significant presence of extractives in *L. alata*, which may play a crucial role in its durability and resistance to corrosion when exposed to harsh outdoor conditions. These abundant extractives are probably associated with the wood’s hardness and corrosion resistance. Further examinations of the chemical components within the extractives acting as quenchants would be of significant interest in subsequent investigations. In summary, the findings of this study indicate the extremely faint autofluorescence of *L. alata* and the presence of abundant extractives that may contribute to its unique properties. Subsequent research should focus on the characterization of these chemical components within the extractives and their role in fluorescence quenching phenomena.

## Figures and Tables

**Figure 1 polymers-15-03269-f001:**
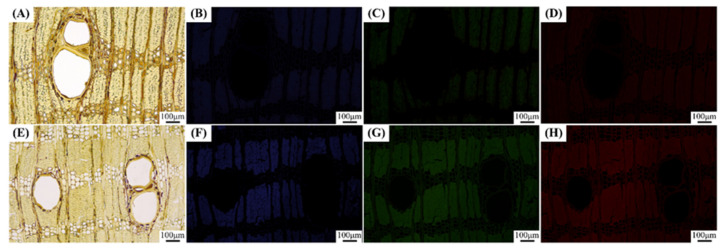
Microscopic images observed in bright field and fluorescence on cross section of the heartwood (**A**–**D**) and sapwood (**E**–**H**) of *Lophira alata*. Observation modes: visible light (**A**,**E**), UV light excitation (**B**,**F**), blue light excitation (**C**,**G**), and green light excitation (**D**,**H**), respectively. Scale bars = 100 μm.

**Figure 2 polymers-15-03269-f002:**
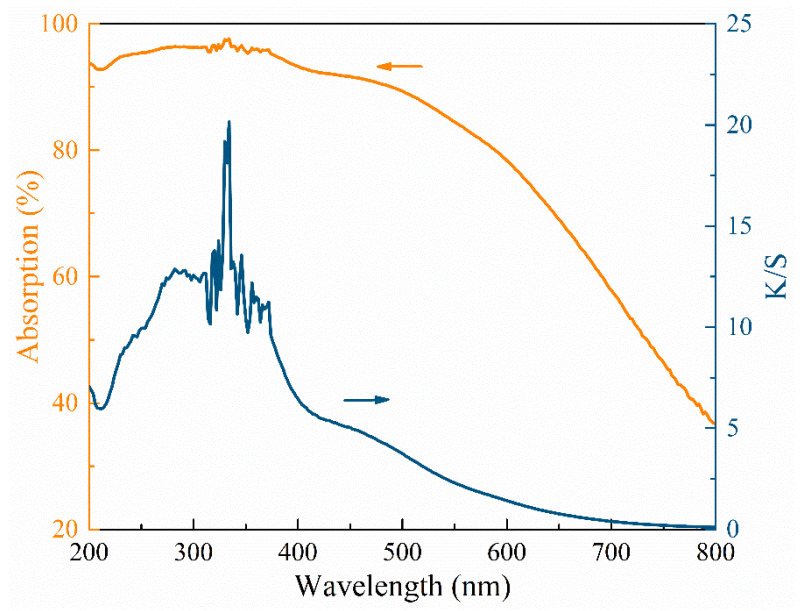
The absorption (left) and *K/S* (right) spectra of *Lophira alata*.

**Figure 3 polymers-15-03269-f003:**
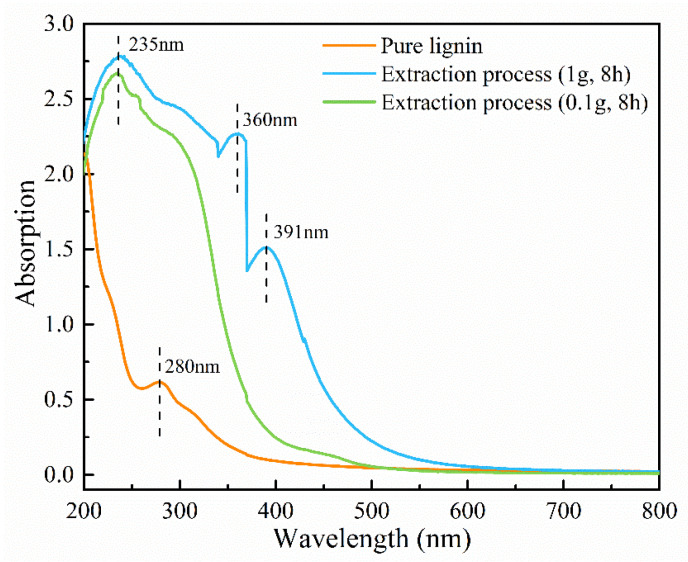
UV–Vis absorption spectra of extracted solutions and commercial pure lignin.

**Figure 4 polymers-15-03269-f004:**
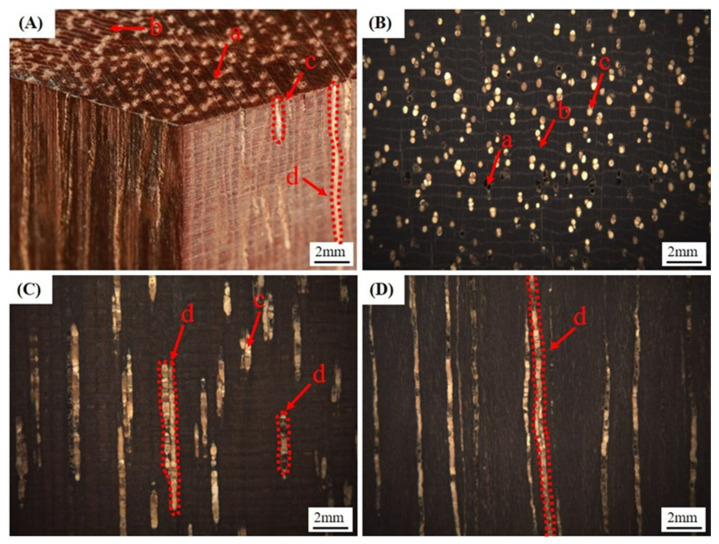
The macrostructure photos of *Lophira alata* in stereo image (**A**), cross section (**B**), radial section (**C**), and tangential section (**D**). Vessel pore (a), longitudinal parenchyma (b), extractives (c), vessel channel (d). Scale bars = 2 mm.

**Figure 5 polymers-15-03269-f005:**
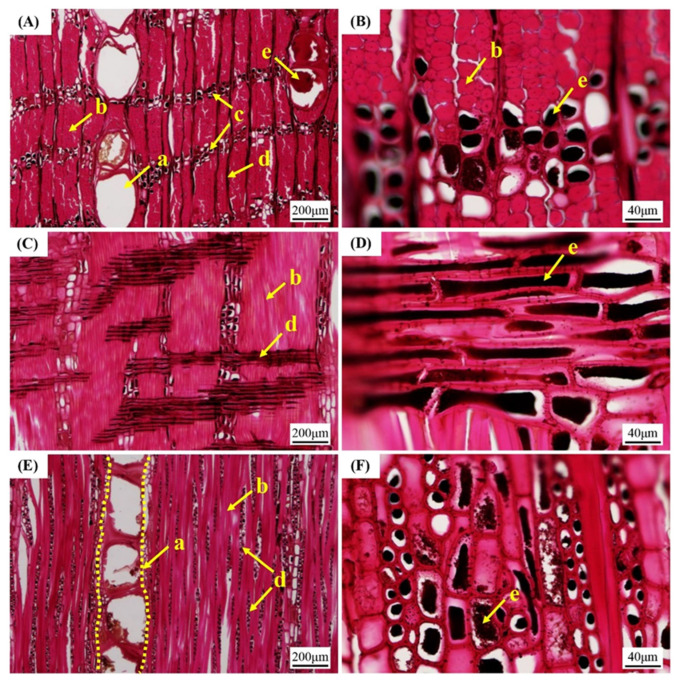
The light microscopy image with safranin staining of *Lophira alata* in cross section (**A**,**B**), radial section (**C**,**D**), and tangential section (**E**,**F**). B, D, and F: the magnified view of fiber, longitudinal parenchyma, and wood ray. Vessel (a), fiber (b), longitudinal parenchyma (c), wood ray (d), extractives (e). Scale bars: A, C, E = 200 μm; B, D, F = 40 μm.

**Figure 6 polymers-15-03269-f006:**
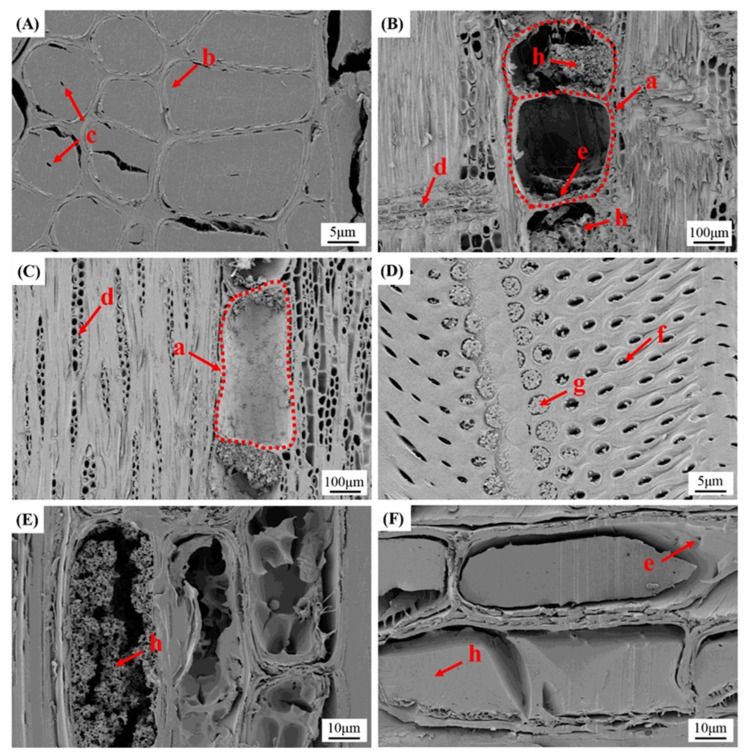
SEM images of *Lophira alata* in cross section (**A**), radial section (**B**), and tangential section (**C**). Magnified view of vessels wall (**D**), longitudinal parenchyma (**E**), and wood ray (**F**) in tangential section. Vessel (a), fiber (b), cell lumen of fiber (c), wood ray (d), perforation plate (e), pits in the vessel wall (f), vestured pitting (g), extractives (h). Scale bars: A, D = 5 μm; B, C = 100 μm; E, F = 10 μm.

**Figure 7 polymers-15-03269-f007:**
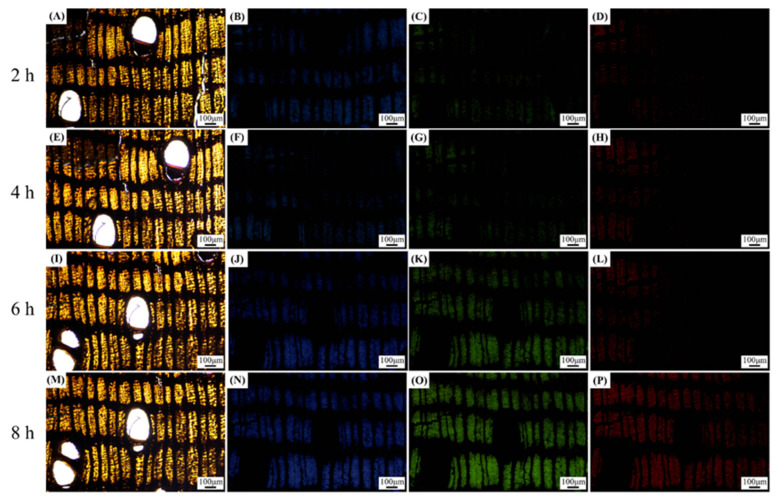
Microscopic images observed in bright field and fluorescence of *Lophira alata* after 2, 4, 6, and 8 h benzene-ethanol extraction, respectively. Observed in visible light (**A**,**E**,**I**,**M**), UV light (**B**,**F**,**J**,**N**), blue light (**C**,**G**,**K**,**O**), green light (**D**,**H**,**L**,**P**). Scale bars = 100 μm.

**Table 1 polymers-15-03269-t001:** Fiber morphology in cross section of *Lophira alata*.

Measurements	Average	Maximum	Minimum	Standard Deviation	Variation Coefficient/%
Double wall thickness/µm	14.6	16.8	12.3	1.0	7.2
Cell lumen diameter/µm	2.4	2.9	1.9	0.3	12.3
Wall-to-lumen ratio	5.9	7.6	5.1	0.7	11.7
Cavity-to-diameter ratio	0.14	0.16	0.12	0.01	9.63

Notes: the number of analyzed fibers is 30.

**Table 2 polymers-15-03269-t002:** Tissue ratios of *Lophira alata*.

Measurements	Average	Maximum	Minimum	Standard Deviation	Variation Coefficient/%
Vessels/%	10.2	13.3	8.6	0.02	15.9
Fibers/%	58.2	61.7	55.8	0.02	3.3
Longitudinal parenchyma/%	14.4	16.2	10.9	0.01	10.2
Rays/%	17.2	18.7	15.7	0.01	6.1

Notes: the number of analyzed images is 12.

## Data Availability

Data generated or analyzed during this study are included in this published article and its Appendix A.

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
