# Peer review of "Improving the Autofluorescence of Lophira alata Woody Cells via the Removal of Extractives"

_polymers, 2023, doi:10.3390/polym15153269_

Round 1
Reviewer 1 Report
Dear authors,
The current paper deals with a very novel paper, especially regarding the domain of lignin. However, at its current state it does not meet the minimum standard that any scietific paper should have.
-The format of the abstract must be reviewed, do not include the name of the different sections in the abstract. The abstract should be plain text in one only paragraph and the ideas should be well-linked and coherent.
-The introduction section is too brief and unclear. In fact, here it must be clearly stated the state of the art of the work, the objetive of the work and the gaps that your work is going to fill, which were not cover by other previous works. This part must be considerably improved.
-In line 71, what do the authors mean by "the samples were softened"
-In line 84, on what basis was the mixture benzene-ethanol selected as solvent for the removal of the extractives.
-What was the objective of the hydrothermal carbonization? This should be better described.
-The macrostructure analized in section 2.4 was based in any standard?
-It is not clear how the results showed in figure 3, are interrelated with the previous one in figure 2.
-It would be easier for the reader if sections 3. Results and 4. Discussion would be merged in the same one.
-I was missing some chemical quantitative or semiquantitative techniques of analysis (FTIR, NMR, GC-MS) to measure the amount of extractives or the changes in the structure after removing the extractives.
-In general the presentation of results and the discussion should improved and worked more in detailed.
-The conclusion were too summarized as well.
In this state I do not consider that the work can be published in this journal.
-
In general terms the quality of the english is good. There are not many grammar, spelling mistakes and lack of coherence between the sentences.
Author Response
Dear Editors and Reviewers,
Thanks for your friendly comments and professional suggestions to modify our manuscript entitled Elaborating to autofluorescence-quenching phenomenon of woody cell in Lophira alata (Manuscript ID: polymers-2423476). These comments are particularly valuable and helpful for improving the academic rigor of our article. Based on your suggestions and requests, we have carefully considered and corrected the modifications in the revised manuscript. The ideas of the revisions are listed as following (All revisions have been noted by the “Track Changes” of Word software in our revised manuscript):
Response to Reviewer 1#:
Reviewer 1#:
1-The format of the abstract must be reviewed, do not include the name of the different sections in the abstract. The abstract should be plain text in one only paragraph and the ideas should be well-linked and coherent.
The authors’ answer: We are very grateful for reviewer’s careful reading. We have revised the format of the abstract to address your concerns and hope that it is now clearer.
2-The introduction section is too brief and unclear. In fact, here it must be clearly stated the state of the art of the work, the objetive of the work and the gaps that your work is going to fill, which were not cover by other previous works. This part must be considerably improved.
The authors’ answer: We appreciate the reviewer for this recommendation. We have revised and improved the introduction section, adding the objectives of the work and the gaps that the work is going to fill.
3-In line 71, what do the authors mean by "the samples were softened"
The authors’ answer: Thank you for this valuable feedback. In a microscopy, wood anatomy is inspected by observing wood slices. Since natural Lophira alata is very hard, it is difficult to directly slice. It is a conventional processing method to soften the hard wood. In the study, the samples have been softened in the boillnig water. In the bibliography, approximately 1 cm3 sectioning blocks have been boiled in water to soften, as following: Gasson et al. (2010), MacLachlan and Gasson (2010), and so on.
Gasson P, Miller R, Stekel DJ, Whinder F, ZiemiÅ„ska K. Wood identification of Dalbergia nigra (CITES Appendix I) using quantitative wood anatomy, principal components analysis and naïve Bayes classification, Ann. Bot. 2010; 105 (1): 45-56.
Maclachlan I R, Gasson P. PCA of cites listed pterocarpus santalinus (leguminosae) wood. IAWA journal, 2010; 31(2): 121-138.
4-In line 84, on what basis was the mixture benzene-ethanol selected as solvent for the removal of the extractives.
The authors’ answer: We sincerely appreciate the significant suggestions. From literatures, a solvent mixture of benzene and ethanol is widely recognized as a standard method for removing most wood extractives. The benzene-ethanol mixture exhibits excellent solubility, effectively dissolving various components present in wood, including fats, resins, and terpenes, which are the major constituents of the extractives from Lophira alata (stated in lines 133 of original manuscript).
5-What was the objective of the hydrothermal carbonization? This should be better described.
The authors’ answer: Hydrothermal carbonization treatment can convert organic matter in wood powder into dissolved organic matter, which may contain compounds with absorption properties, such as organic acids, phenols, aldehydes, etc. By measuring the UV-Vis absorption spectra of the supernatant, the absorbance of these organic compounds at different wavelengths can be obtained, and thus the types and concentrations of the compounds in the products can be inferred (stated in lines 136 of original manuscript).
6-The macrostructure analized in section 2.4 was based in any standard?
The authors’ answer: The main references are standards related to macroscopic observation of wood. The most important of these are ISO 3823-1:2007 "Wood - Physical and mechanical tests - Part 1: Methods for determining properties" and ISO 3823-2:2017 "Wood - -Physical and mechanical tests - Part 2: Methods for determining structural properties.
7-It is not clear how the results showed in figure 3, are interrelated with the previous one in figure 2.
The authors’ answer: We are so grateful for your kind question. The sample for the diffuse reflection spectra in Figure 2 was wood powder, it confirms the extremely high light absorption capacity of the wood. And Figure 3, by measuring the UV-Vis absorption spectra of the supernatant after hydrothermal carbonization, we would like to further verify whether the high light absorption capacity of the wood was the internal extractives component playing a major role.
8-It would be easier for the reader if sections 3. Results and 4. Discussion would be merged in the same one.
The authors’ answer: We appreciate the reviewer for this recommendation. The section 4 is mainly a discussion of the previous results, describing the possible factors affecting the fluorescence of the material, closely linking the results of each test in the section 3, and also designing extraction experiments to verify the effect of extractives on lignin fluorescence. It is concluded that a large amounts of extractives in wood can quench the fluorescence of lignin. So I think it would be better not to merge them together.
9-I was missing some chemical quantitative or semiquantitative techniques of analysis (FTIR, NMR, GC-MS) to measure the amount of extractives or the changes in the structure after removing the extractives.
The authors’ answer: We thank the reviewer for pointing this out. Due to some current limitations, these chemical quantitative or semiquantitative techniques of analysis will be continued in subsequent studies in our group.
10-In general the presentation of results and the discussion should improved and worked more in detailed.
The authors’ answer: Thank you for your valuable feedback and suggestions regarding the presentation of results and the discussion in our manuscript. We appreciate your thorough review and have carefully considered your comments. In response to your concerns, we have made significant revisions to enhance the clarity and detail of these sections. Furthermore, we have expanded upon the interpretation and discussion of the results, addressing the underlying mechanisms and implications of our findings. At the same time, we consulted the relevant literature, and also acknowledged any differences or limitations of the current work, which will continue to be improved in the next work of our group.
11-The conclusion were too summarized as well.
The authors’ answer: We appreciate the reviewer for this recommendation. As you mentioned, we have revised and improved the conclusion, and hopefully it is now better presented.
Once again, we sincerely appreciate your thoughtful evaluation of our manuscript and your valuable suggestions. We are grateful for the opportunity to improve our work based on your insightful comments. Should you have any further suggestions or concerns, please do not hesitate to let us know.
Thank you for your time and consideration.
Yours sincerely,
Dr. Jinbo HU
Reviewer 2 Report
The authors set out to prove that after removing some extractives the autofluorescence of Azobe wood cells improved. For an extremely simply study, the experimental design and presentation has to be meticulous.
1) Experimental design: Extractives are the main focus of this study. In figures 4, 5, and 6, the location and extent of extractives has been identified. But how to prove to a non-expert that these trapped structures are extractives? The authors should have compared the raw and benzene-ethanol extracted wood pieces (via optical and SEM microscopy) to clearly demonstrate the removal of "extractives".
2) Experimental design: Sampling is another important factor to prove that the observed changes are universal and that the authors carefully selected a representative sample. Therefore, please show how a "representative sample" was collected.
E.g. Some biomass chemists employ rotary sample divider to select a representative sample.
3) Method description: In section 2.3, please provide details of (a) ratio of benzene to ethanol; (b) ratio of solvent volume to wood powder weight; (c) extraction temperature for the first 8 hours; (d) solid-to-liquid ratio of commercial lignin sample.
What is thermal carbonization and why was this procedure implemented?
Did the commercial lignin dissolve completely in the benzene-alcohol mixture? Did you subject the commercial lignin solution to the same thermal carbonization procedure and then filtered?
4) Results: Figure captions need revision as follows:
For all figures, give the full scientific name of Azobe wood. The figure captions should be self-sufficient and a reader must be able to understand the whole study by just reading the figures and captions.
Figure 1: Clearly state the kind of microscopy (i.e., fluorescence) used.
Is Figure 2 necessary, its meaning is not clear? It could be presented in supplementary file.
Figure 3: Please mention what kind of extraction chemicals were used, i.e., benzene-ethanol.
Figure 4: Clearly state the type of microscopy (optical) and the dye (safranin) used.
5) Results: It is unclear how many sample were analyzed to arrive at the data presented in Tables 1 and 2. Is it even possible to determine how many observations would be sufficient to capture the whole truth, 100 or 200? The authors have to carefully consider this dilemma.
6) Discussion: In sentence 250, you mention that the presence of lignin was observed in the fiber cell wall. Surely you do not mean the brown segments in Figure 4. Despite your claim in Lines 250-251 ("At the same time, the presence of lignin in the fiber cell walls suggest"), the presence of lignin was not highlighted while discussing the microscopy images.
7) Experimental design: Why pick benzene-ethanol solvent combination to remove the extractives? Is it based on previous literature? Aren't hexanes better to remove waxes and fatty acids (e.g. https://doi.org/10.1007/s42452-021-04873-1).
8) Introduction: Provide the reported chemical composition of Azobe wood in the introduction.
9) Results: Based on previous reports (of total extractives content in Azobe wood), try to gravimetrically estimate how much extractives were removed from the wood powders after 8 h. Similarly, compare the efficiency of extractives removal on a 0.1 g scale versus 1 g scale.
1) The English language use is convoluted and often confusing. Please simplify your message.
For example, Lines 46-47, "The microstructure of Azobe wood is occasionally performed during a fluorescence observation by ourself, which is shown in Fig. 1", make no sense. It should be simplified as follows: "The microstructure of Azobe wood is occasionally observed using fluorescence microscopy as shown in Fig.1".
2) Please check spelling.
E.g. in Line 70, it is wood "blocks" not blacks.
Author Response
Dear Editors and Reviewers,
Thanks for your friendly comments and professional suggestions to modify our manuscript entitled Elaborating to autofluorescence-quenching phenomenon of woody cell in Lophira alata (Manuscript ID: polymers-2423476). These comments are particularly valuable and helpful for improving the academic rigor of our article. Based on your suggestions and requests, we have carefully considered and corrected the modifications in the revised manuscript. The ideas of the revisions are listed as following (All revisions have been noted by the “Track Changes” of Word software in our revised manuscript):
Response to Reviewer 2#:
Reviewer 2#:
1-The authors set out to prove that after removing some extractives the autofluorescence of Azobe wood cells improved. For an extremely simply study, the experimental design and presentation has to be meticulous.
1-Experimental design: Extractives are the main focus of this study. In figures 4, 5, and 6, the location and extent of extractives has been identified. But how to prove to a non-expert that these trapped structures are extractives? The authors should have compared the raw and benzene-ethanol extracted wood pieces (via optical and SEM microscopy) to clearly demonstrate the removal of "extractives".
The authors’ answer: The transverse and longitudinal parenchyma cells within the wood can serve as water and inorganic salt transport channels, maintaining metabolic processes and storing nutrients. In this process, extractives are slowly deposited in the cell walls or filled in the cell lumen and some cell tissues. On this basis, the black components in the cell lumen (Figure -5 B/D/F) wood rays and parenchyma tissues under optical microscopy, as well as the intracellular material in the cell lumen (Figures 6-D/E/F), can be identified as extractives. We could easily observed the lignin autofluorescence in the benzene-ethanol extracted wood pieces by fluorescence, and the partial removal of extractives made the lignin autofluorescence more obvious, which is a simpler and more intuitive method. In addition, the benzene-ethanol extracted wood pieces were not stained, so it was difficult to observe lignin directly.
2) Experimental design: Sampling is another important factor to prove that the observed changes are universal and that the authors carefully selected a representative sample. Therefore, please show how a "representative sample" was collected.
E.g. Some biomass chemists employ rotary sample divider to select a representative sample.
The authors’ answer: Thank you for your valuable opinions. Firstly, wood discs about 5 cm thick were collected on trees of 1.2 m height and then stored for more than 6 months to adjust the moisture content to 12%. Next, samples were selected in the medium heartwood for being the most uniform ones in color.
3) Method description: In section 2.3, please provide details of (a) ratio of benzene to ethanol; (b) ratio of solvent volume to wood powder weight; (c) extraction temperature for the first 8 hours; (d) solid-to-liquid ratio of commercial lignin sample.
What is thermal carbonization and why was this procedure implemented?
Did the commercial lignin dissolve completely in the benzene-alcohol mixture? Did you subject the commercial lignin solution to the same thermal carbonization procedure and then filtered?
The authors’ answer: Here, we added experimental details: (a) The ratio of benzene to 95% alcohol is 2:1. (b) The ratio of solvent volume to wood powder weight is 30:1 (mL: g). (c) The extraction temperature for the first 8 hours is in a water bath temperature of ~78℃. According to the principle of keeping the benzene alcohol mixture boiling during the extraction process, the degree of boiling is appropriate to circulate not less than 4 times per hour, thus determining the temperature of 78℃. (d) The solid-to-liquid ratio of commercial lignin sample is 30:1 (mL: g).
Hydrothermal carbonization treatment converts organic matter in wood powder into dissolved organic matter, which may contain compounds with absorption properties, such as organic acids, phenols, aldehydes, etc. Then by measuring the UV-Vis absorption spectra of the supernatant, the absorbance of these organic compounds at different wavelengths can be obtained, and thus the types and concentrations of the compounds in the products can be inferred.
Pure commercial lignin is largely dissolved in benzene-alcohol mixtures. Yes, I subject the commercial lignin solution to the same thermal carbonization procedure and then filtered.
4) Results: Figure captions need revision as follows:
For all figures, give the full scientific name of Azobe wood. The figure captions should be self-sufficient and a reader must be able to understand the whole study by just reading the figures and captions.
Figure 1: Clearly state the kind of microscopy (i.e., fluorescence) used.
Is Figure 2 necessary, its meaning is not clear? It could be presented in supplementary file.
Figure 3: Please mention what kind of extraction chemicals were used, i.e., benzene-ethanol.
Figure 4: Clearly state the type of microscopy (optical) and the dye (safranin) used.
The authors’ answer: Thank you for this valuable feedback. For all figures, the full scientific name of Azobe wood has been added.
In Figure 1 we added the state of fluorescence microscopy used.
Figure 2: Certainly Figure 2 is necessary. The diffuse reflectance spectra in Figure 2 provides information on the absorption and reflection of wood at different wavelengths of light. From Figure 2 we know that Azobe wood has a strong absorption capacity for light, especially in the wavelength range below 500nm. And the abundant extractives may affect its color and light absorption ability.
Figure 3: The extraction chemicals used were mixed solutions of benzene (C6H6) and 95% alcohol (with the ratio of benzene to alcohol is 2:1). (Added in line 107)
Figure 4: The type of microscopy is macroscopic stereo microscopy (MZS0745), no dye was used in Figure 4. In Figure 5 the samples were stained with 1% safranine O solution.
5) Results: It is unclear how many sample were analyzed to arrive at the data presented in Tables 1 and 2. Is it even possible to determine how many observations would be sufficient to capture the whole truth, 100 or 200? The authors have to carefully consider this dilemma.
The authors’ answer: We thank reviewer for the valuable suggestions. After extensive preliminary observations, we selected 30 morphologically homogeneous fibers, and 12 cross-sectional microscopic observation images for tissue ratio measurements, which finally resulted in the data in Tables 1 and 2 (stated in lines 151 and 153 of original manuscript). of original manuscript
6) Discussion: In sentence 250, you mention that the presence of lignin was observed in the fiber cell wall. Surely you do not mean the brown segments in Figure 4. Despite your claim in Lines 250-251 ("At the same time, the presence of lignin in the fiber cell walls suggest"), the presence of lignin was not highlighted while discussing the microscopy images.
The authors’ answer: Thank you for your valuable opinions. According to the literature, lignin is contained in the cell wall of Lophira alata, which can be verified by the gradual appearance of lignin autofluorescence after the cells were extracted in Figure 7. The presence of lignin was not highlighted while discussing the microscopy images, because the lignin in the cell wall cannot be directly observed under an optical microscopy, it needs to be observed by fluorescence or other specific stains.
7) Experimental design: Why pick benzene-ethanol solvent combination to remove the extractives? Is it based on previous literature? Aren't hexanes better to remove waxes and fatty acids (e.g. https://doi.org/10.1007/s42452-021-04873-1).
The authors’ answer: We thank reviewer for the valuable suggestions. From literatures, a solvent mixture of benzene and ethanol is widely recognized as a standard method for removing most wood extractives. It has been reported that the main components of Lophira alata extractives mainly include polyphenols, tannins, flavonoids, alkaloids, etc. Compared to hexane, benzene-alcohol is a polar solvent and has better solubility for many polar compounds. Benzene-alcohol has certain antioxidant properties, which help to protect the compounds in the extract from oxidative damage and thus better retain their original properties. Therefore, benzyl alcohol extraction was chosen in this experiment.(e.g. https://doi.org/10.1515/jbcpp-2014-0096, https://doi.org/10.9734/ejmp/2020/v31i1130295 )
8) Introduction: Provide the reported chemical composition of Azobe wood in the introduction.
The authors’ answer: We appreciate the reviewer for this recommendation. As you mentioned, we have added information of the composition of the Lophira alata extracts in the introduction. “It has been reported that L. alata wood has rich extractives within it, and the main components include polyphenols, tannins, flavonoids, alkaloids, etc.” (state in lines 70).
9) Results: Based on previous reports (of total extractives content in Azobe wood), try to gravimetrically estimate how much extractives were removed from the wood powders after 8 h. Similarly, compare the efficiency of extractives removal on a 0.1 g scale versus 1 g scale.
The authors’ answer: Thank you for your valuable feedback and suggestions regarding the estimation of extractives removal from the wood powders and the comparison of efficiency on different scales.It is very important that to present an estimation of the extractives removed from the wood powders after 8 hours of treatment. The potential application from the study have been conceived and arranged to look for a optical material. In the next few time, the amount of extractives removed from wood slices could be examned by some chemical methods. In the manuscript, the different treatments of benzene-ethanol extraction have been presented the possible reason of autofluorescence-quenching phenomenon. It is very interesting to hamper the lignin autofluorescence by some natural organics in the plants. Then, the manuscript is going to submit to journal by the primary researcher, who could get an encouragement to explore the scientific investigation, focusing on the plant study and its potential application.
Comments on the Quality of English Language
1) The English language use is convoluted and often confusing. Please simplify your message.
For example, Lines 46-47, "The microstructure of Azobe wood is occasionally performed during a fluorescence observation by ourself, which is shown in Fig. 1", make no sense. It should be simplified as follows: "The microstructure of Azobe wood is occasionally observed using fluorescence microscopy as shown in Fig.1".
The authors’ answer: We are sorry for the English language use and grammatical problems and have correct them based on your suggestions. We have sought out a help from the native English speakers to polish the manuscript.
2) Please check spelling.d
E.g. in Line 70, it is wood "blocks" not blacks.
The authors’ answer: We are very grateful for reviewer’s careful reading. Ok, the "blocks" in the original text has been revised, and we have further checked the spelling.
Once again, we sincerely appreciate your thoughtful evaluation of our manuscript and your valuable suggestions. We are grateful for the opportunity to improve our work based on your insightful comments. Should you have any further suggestions or concerns, please do not hesitate to let us know.
Thank you for your time and consideration.
Yours sincerely,
Dr. Jinbo HU
Round 2
Reviewer 1 Report
Dear author,
I gladly saw that the authors have significantly improved their manuscript and that they have taken into account my considerations. I now considered it may be considered for publication, but before some minor points must be still corrected and improved:
-Authors should briefly review the grammar and spelling through the article to correct some remaining mistakes.
-The standards mentioned in the answers to my concern should be included in the section 2.4 of the manuscript.
-Thank you to the authors for clarifying the question concerning figures 2 and 3. Now it is clearer to me. Nevertheless, to avoid this type of misconception I would suggest that authors would do the conversion of reflectance to absorbance for figure 2. Thereby, the axis of figures 2 and 3 would be the same and the comparison and relationship between both figures would be more explicit.
Should these point be considered, it might be published in this journal.
The article is written in a way that can be understood by the majority of readers. Nevertheless, it should be still reviewed to correct some mistakes concerning English gramma and spelling.
Author Response
Dear Editors and Reviewers,
Thanks for your friendly comments and professional suggestions to modify our manuscript entitled Elaborating to autofluorescence-quenching phenomenon of woody cell in Lophira alata (Manuscript ID: polymers-2423476). We have fully revised our manuscript and have addressed all of reviewer’s comments. The detailed revisions are listed below and highlighted in the revised manuscript with yellow background.
Reviewer 1:
1-Authors should briefly review the grammar and spelling through the article to correct some remaining mistakes.
The authors’ answer: Thank you for your valuable feedback on the manuscript. We have thoroughly reviewed the grammar and spelling throughout the article and have made the necessary corrections to address any remaining mistakes. We appreciate your attention to detail and guidance in improving the language quality of the manuscript. At same time, the manuscript have been especially inspected by Prof. Xiaodong (Alice) Wang at Laval University.
2-The standards mentioned in the answers to my concern should be included in the section 2.4 of the manuscript.
The authors’ answer: Ok, the standards mentioned in the answers were added in the section 2.4 of the manuscript. (could be seen in reference 19: ISO 3129:2019, titled "Wood - Sampling methods and general requirements for physical and mechanical tests.)
3-Thank you to the authors for clarifying the question concerning figures 2 and 3. Now it is clearer to me. Nevertheless, to avoid this type of misconception I would suggest that authors would do the conversion of reflectance to absorbance for figure 2. Thereby, the axis of figures 2 and 3 would be the same and the comparison and relationship between both figures would be more explicit.
The authors’ answer: Thank you for providing your valuable feedback and suggestion regarding figures 2 and 3. We appreciate your input, and we agree that converting reflectance to absorbance for figure 2 would make the comparison and relationship between both figures more explicit. Based on your suggestion, we have carefully considered your feedback and made the necessary modifications to the figures (could be seen in figure 2). In addition, in order to maintain the coordination of the overall graphics, we have also made necessary adjustments to Figure 3.
By converting the reflectance values to absorbance, the axis of figures 2 and 3 will now be the same, ensuring a clearer and more direct comparison. This modification enhances the visual representation of the data, enabling a better understanding of the relationship between the two figures.
Once again, we sincerely thank you for your insightful feedback and for helping us improve the presentation of our research. Please let me know if there are any further revisions or modifications needed.
Thank you for your time and consideration.
Yours sincerely,
Dr. Jinbo HU
Reviewer 2 Report
1) The current title is inappropriate, because the authors are not elaborating the mechanism of autofluorescence quenching. Instead the title could be altered as follows, "Improving the autofluorescence of Lophira alata woody cells via extractives removal".
2) In the introduction, please provide the chemical composition of Lophira alata wood, i.e., cellulose, hemicellulose, lignin, total extractives, and ash – as weight percentage of whole biomass. This information should be readily available in previous literature.
3) In figure 5 caption, please mention that it is a "light microscopy image with safranin staining".
4) Please mention the number of samples analyzed as a footnote in Tables 1 and 2.
Please review the English language structure and accuracy while proofreading.
Author Response
Dear Editors and Reviewers,
Thanks for your friendly comments and professional suggestions to modify our manuscript entitled Elaborating to autofluorescence-quenching phenomenon of woody cell in Lophira alata (Manuscript ID: polymers-2423476). We have fully revised our manuscript and have addressed all of reviewer’s comments. The detailed revisions are listed below and highlighted in the revised manuscript with yellow background.
1) The current title is inappropriate, because the authors are not elaborating the mechanism of autofluorescence quenching. Instead the title could be altered as follows, "Improving the autofluorescence of Lophira alata woody cells via extractives removal".
The authors’ answer: Thank you for your valuable feedback regarding the title of the manuscript. We appreciate your suggestion to alter the title to better reflect the content of our research. After careful consideration, we would like to get your suggestions. Based on your recommendation, we have revised the title as follows: "Improving the autofluorescence of Lophira alata woody cells via extractives removal." We sincerely appreciate your input, as it has helped us enhance the clarity and relevance of our paper's title.
2) In the introduction, please provide the chemical composition of Lophira alata wood, i.e., cellulose, hemicellulose, lignin, total extractives, and ash – as weight percentage of whole biomass. This information should be readily available in previous literature.
The authors’ answer: Thank you for bringing up the suggestions regarding the chemical composition of Lophira alata wood. The chemical compositions of Azobe wood should be investigated to assist this study. Furthermore, the extractives should be progressively explored because it is crucial to the autofluorescence-quenching phenomenon. We have thoroughly sought out the available literature, and unfortunately, we cannot find specific information regarding the weight percentages of cellulose, hemicellulose, lignin, total extractives, and ash in Lophira alata wood.
However, we did come across a relevant piece of information regarding heartwood of Lophira alata. According to the literature, the heartwood of Lophira alata contains unusually high levels of Klason residues, approximately 40.0% by weight, which generally correspond to the lignin content of wood[1]. In addition, the extractives have been explained that it is higher than some European woods[2-7]. The references are listed as following:
[1] Lange, W. , & Faix, O. (1999). Lignin-polyphenol interaction in azobe (lophira alata) heartwood. a study on milled wood lignin (MWL) and klason residues. Holzforschung, 53(5), 519-524.
[2] Sen, S. Sivrikaya, H. Yalçin, M. (2009). Natural durability of heartwoods from European and tropical africa trees exposed to marine conditions. Afr. J. Biotechnol. 8 (18), 4425-4432.
[3] Palanti, S. Feci, E. Anichini, M. (2015). Comparison between four tropical wood species for their resistance to marine borers (Teredo spp and Limnoria spp) in the Strait of Messina. International Biodeterioration & Biodegradation, 104, 472-476, http://dx.doi.org/10.1016/j.ibiod.2015.07.013
[4] Edoun, F. L. E. Tchuente, B. R. T. Dibacto, R. E. K. et al. (2020) Phytochemical screening and antioxidant potential of aqueous extracts of Millettia laurenti, Lophira alata and Milicia excelsa, commonly used in the Cameroonian pharmacopoeia. European Journal of Medicinal Plants, 31(11), 11-23.
[5] Mouafo, H. T. Tchuenchieu, A. D. K. Nguedjo, M. W. et al. (2021) In vitro antimicrobial activity of Millettia laurentii De Wild and Lophira alata Banks ex CF Gaertn on selected foodborne pathogens associated to gastroenteritis. Heliyon, 7(4). http://dx.doi.org/10.1016/j.heliyon.2021.e06830
[6] Van, d. K. J. W. G. & Blass, H. J. (2005). Mechanical properties of azobe (lophira alata). Holz als Roh- und Werkstoff, 63,1-10, http://dx.doi.org/10.1007/s00107-004-0533-7
[7] Sandhaas, C. & Van, d. K. J. G. (2017). Strength and stiffness of timber joints with very high strength steel dowels. Engineering Structures, 131, 394-404, http://dx.doi.org/10.1016/j.engstruct.2016.10.046
3) In figure 5 caption, please mention that it is a "light microscopy image with safranin staining".
The authors’ answer: Thank you for providing your feedback regarding the caption of figure 5. We appreciate your suggestion to mention that the image is a "light microscopy image with safranin staining." Based on your input, we have updated the caption of figure 5 to include this information.
4) Please mention the number of samples analyzed as a footnote in Tables 1 and 2.
The authors’ answer: Thank you for your valuable opinions. Based on your recommendation, the number of samples analyzed were added under Tables 1 and 2.
Comments on the Quality of English Language
Please review the English language structure and accuracy while proofreading.
The authors’ answer: We would like to express plenty of gratitude for your valuable feedback and review of the English language in the manuscript. We have thoroughly checked and revised the English language as per your requirements. On this occasion, the manuscript have been especially inspected by Prof. Xiaodong (Alice) Wang at Laval University, Canada.
Once again, we sincerely thank you for your insightful feedback and for helping us improve the presentation of our research. Please let me know if there are any further revisions or modifications needed.
Thank you for your time and consideration.
Yours sincerely,
Dr. Jinbo HU